# Physical, Metabolic, and Microbial Rumen Development in Goat Kids: A Review on the Challenges and Strategies of Early Weaning

**DOI:** 10.3390/ani13152420

**Published:** 2023-07-26

**Authors:** Mahmoud M. Abdelsattar, Wei Zhao, Atef M. Saleem, Ahmed E. Kholif, Einar Vargas-Bello-Pérez, Naifeng Zhang

**Affiliations:** 1Key Laboratory of Feed Biotechnology of the Ministry of Agriculture and Rural Affairs, Institute of Feed Research of Chinese Academy of Agricultural Sciences, Beijing 100081, China; m.m.abdelsattar@agr.svu.edu.eg (M.M.A.); zhaowei45@nwafu.edu.cn (W.Z.); 2Department of Animal and Poultry Production, Faculty of Agriculture, South Valley University, Qena 83523, Egypt; a.seliem@agr.svu.edu.eg; 3College of Animal Science and Technology, Northwest A&F University, Xianyang 712100, China; 4Department of Dairy Science, National Research Centre, Giza 12622, Egypt; ae_kholif@live.com; 5Department of Animal Sciences, School of Agriculture, Policy and Development, University of Reading, P.O. Box 237, Earley Gate, Reading RG6 6EU, UK; e.vargasbelloperez@reading.ac.uk; 6Facultad de Zootecnia y Ecología, Universidad Autónoma de Chihuahua, Periférico R. Aldama Km 1, Chihuahua 31031, Mexico

**Keywords:** goat kids, rumen development, early weaning, weaning stress

## Abstract

**Simple Summary:**

The rumen development process that includes the physical, metabolic, and microbial development from birth to postweaning stage in goat kids was reviewed. Moreover, the role of different rearing systems on rumen development was extensively elaborated, especially those related to early weaning strategies. This review emphasized strong structural and functional changes related to rumen development in newborn goats, which often occurs at weaning.

**Abstract:**

The digestive system of newborn ruminant functions is similar to monogastric animals, and therefore milk flows into the abomasum instead of rumen for digestion. The rumen undergoes tremendous changes over time in terms of structure, function, and microbiome. These changes contribute to the smooth transition from the dependence on liquid diets to solid diets. Goat kids are usually separated at early ages from their dams in commercial intensive systems. The separation from dams minimizes the transfer of microbiota from dams to newborns. In this review, understanding how weaning times and methodologies could affect the normal development and growth of newborn goats may facilitate the development of new feeding strategies to control stress in further studies.

## 1. Introduction

The demand for safe and healthy goat milk and meat products is increasing with the growth of population and income increases [1]. With about one billion stock and 200 breeds, goats produce milk with high nutritional value, meat with high protein and low cholesterol, and wool of high quality [2]. Accordingly, goats play an important role in agricultural economics and ecological niches [3]. Compared with large ruminants, goats have a shorter production cycle, higher development rate, and greater environmental adaptation, and can cope with different stressors, harsh environments, and diseases [4]. Compared to sheep, goats are an intermediate browser preferring trees and shrubs [5,6].

As small ruminant species, goat kids are born with deficient physical, metabolic, and microbial rumen development [7]. The rumen is the point of contact between the host and the nutrients consumed [8], where most nutrient digestion and metabolism occur [9]. The process of rumen development, including morphology, metabolic function, and microbial colonization, is a temporal and successional process during the early stage of life [10]. The microbiota colonization process has been defined as a co-evolutionary process because of the interplay between the host and the microbes [11]. The microbiota population is established through consecutive waves in which microbial populations converge and a more stable population structure emerges [12]. Ingested microorganisms colonize and establish in a definite and progressive sequence in juvenile ruminants’ rumens, according to several studies [13,14,15]. The early colonization of rumen microorganisms influences the maturation of rumen function and affects the rumen microbiome composition throughout life [16,17,18], as well as has a long-term effect on the health of adult ruminants and animal production [19]. In newborn calves, colonization of the stomach with symbiotic and commensal bacteria has been shown to be critical for the establishment and maintenance of the neonate’s immune system [20]. In addition, methanogen colonization in the rumen’s early life contributes to the knowledge of the modulation of the rumen microbiome, methane mitigation, and production efficiency [21].

Several goat rearing practices are practiced including extensive rearing, semi-intensive, and intensive. In traditional extensive rearing systems, goats are continuously kept in the open field compared to goats under intensive rearing systems, where they are kept indoors and stall-fed. In the traditional rearing system, the long lactation period and high mortality rate of goat kids have caused serious shortages of productive wealth sources, which restricts the development of goat production. Therefore, early weaning technology has become a common practice for goat breeding in many countries because it can reduce the production costs and increase the birth rate of the dam and the marketable milk as well as the adaptation of young ruminants to the new solid diet [22,23,24,25]. However, compared to the traditional weaning, early weaning may cause a stronger postweaning stress due to the immature rumen function [7]. Weaning stress causes impairment of the digestive system, growth retardation, diarrhea, and lower respiratory tract infection, leading to the increase in mortality rates [23,26,27,28]. Sudden and early weaning can disturb the microbial composition in the gastrointestinal tract, change the metabolic function and the construction of the intestine, affect the newborn performance, and increase the rate of neonatal diarrhea and mortality [26,27]. Consequently, in intensive feeding systems, early weaning requires strict management practices and deep attention [29]. The present review discusses available data that could be useful for the development of feeding strategies to manipulate rumen development and mitigate early weaning stress and reduce high mortality rates.

## 2. Materials and Methods

The information search focused on studies reporting physical, metabolic, and microbial rumen development in goat kids in relation to weaning. The literature search was conducted (up to June 2023) using PubMed, Science Direct, Web of Science, Scopus, and Google Scholar databases. It summarized only peer-reviewed papers written in English language and findings from recent years concerning the rumen development in goats from neonatal to postweaning stage, whilst conference and congress contributions were excluded. While an effort was made to focus on research carried out in goat kids, we chose to include studies in other ruminant species pertinent to the discussion.

## 3. Rumen Development of Goats

### 3.1. Rumen Morphology and Rumination Function

The rumen mucosa structure, including size and shape of papilla, has a significant role in nutrient digestion or absorption [30]. The main factors affecting rumen epithelial development are diets and age [10,31,32]. At birth, the rumen papillae growth is minimal in goat kids [10] and calves [33]. The rumen is not engaged in the digestion of plant material or milk during the suckling stage [34,35,36]. This is because a reflex mechanism closes the esophageal groove directing the colostrum and milk flow to the abomasum for enzymatic digestion [19,37,38].

The transition period starts from 3 to 8 weeks of age when the rumen gradually develops [39] and adapts to the consumption of solid feed [40]. Several studies reported the beneficial effects of solid feed in stimulating rumen development through the early stage of life in ruminants [32,41]. As a result of the introduction of a solid diet, the length and size of the rumen papillae in goat kids rose gradually during the transition period [10,42]. This is supported by Chai et al. [43], who observed that the most important factor that altered rumen microbiota and epithelial gene expression in pre-weaning ruminants was the amount of neutral detergent fiber. It may be due to the solid feed providing butyrate as well as a physical stimulus favoring rumen anatomical development [31,32]. Htoo et al. [44] showed that the large particle size of roughage and the content of fiber increased the rumen wall by physical stimulation, and consequently enhanced rumen motility, muscularization, and rumen volume in goat kids with free access to creep feed with roughage. In addition, Malhi et al. [45] reported that intraruminal infusion of sodium butyrate at 0.3 g/kg of body weight for 28 days in goats improved the rumen papillae size, density, and surface area. Ruminal infusion of butyrate at 0.3 g/kg of body weight in goats increased the full weight of rumen as percent of total stomach weight [46]. In addition, infusion of sodium butyrate at 0.3 g/kg of body weight can improve ruminal epithelial growth by modulating both proliferative and apoptotic genes, i.e., Bax, caspase 3, and caspase 9 [47]. Moreover, sodium butyrate supplementation at 3.2% of diet increased rumen epithelium thickness in growing rams [48]. Accordingly, the transition period should be considered by a goat nutritionist for the adaptation of rumen papillae to the dietary changes.

After sequential dynamic changes of development [10], the rumen development of a mature animal accounts for 60 to 80% of the complex stomach volume [38,49]. It has been reported that the microbial colonization in the rumen of goats occurred earlier and is achieved at 1 month of age, functional achievement is at 2 months, and anatomic development is achieved after 2 months [10]. Meanwhile, the rumen transforms to a fully functional fermenter with capabilities of utilizing fibrous diets [38]. At this stage, goats are able to break down and extract nutrients from tough fibrous plants through a process of microbial fermentation and regurgitation. The mature rumen serves as a fermentation chamber, where bacteria and protozoa break down cellulose and hemicellulose into simple sugars, organic acids, and gases, which allows goats to efficiently extract nutrients from low-quality and less digestible forages, and adapt to grazing on a variety of vegetations [6].

In addition, the development of rumen absorption function in goat kids after birth is a crucial process that enables animals to adapt to solid diets. Initially, the rumen is a small pouch-like structure with a thin lining and lacks the necessary capacity to perform efficient absorption of nutrients. However, the rumen gradually expands due to the induced cell proliferation and differentiation as the kid starts to consume solid feed [50,51]. Papillae can greatly enhance the absorptive capacity of the rumen by increasing its surface area [45,52]. High-fiber diets induce the expression of genes related to short-chain fatty acid absorption in the rumen epithelium of goats [53]. A similar improvement in absorption capacity of the rumen epithelium was observed in sheep [52] and cows [54], even with unchanged absorptive surface area of the rumen papillae [54]. On the other hand, low-fiber diets can cause insufficient rumen development and impede proper nutrient digestion and absorption [35]. Therefore, it is important to carefully evaluate the impact of age and diets in further studies to determine their effectiveness in enhancing the absorptive capacity of the rumen.

Daily, goats spend much time eating and ruminating [55]. Rumination is a complex process involving regurgitation and chewing of previously swallowed feed. Lickliter [56] reported that kids start mouthing soil and grasses during their first week following birth but ruminating is first observed during week 3 of age. Before weaning, kids ruminate for over 3 h/day [25]. After weaning, removal of dietary milk results in a dramatic increase in the time spent for rumination (5.2 h/day) [25], indicating the major roles of age and feeding [25,55,56]. Finally, it has been reported that the adult goats spend 7.2 h/day reaching about one third of their day for rumination [57,58].

### 3.2. Microbiota

#### 3.2.1. The Importance of Rumen Microbes

Various prokaryotic (bacteria and archaea) and eukaryotic (protozoa and fungal) microorganisms live in the rumen and work together to digest and ferment feed [11]. The ruminal microbiota has a symbiotic relationship with the host and is distinguished by its high population density, diversity, and complexity of interactions [21,34]. Rumen microbes have a remarkable ability to ferment and transform plant feed into microbial matter, volatile fatty acids (VFAs), fermentation gases (methane and carbon dioxide), and ammonia, as well as producing heat [43,59,60]. The VFA and microbial proteins provide nutrients for the host’s maintenance and growth [34,61]. The rumen microorganisms, especially bacteria and their end products, are essential for regulating rumen function and nutrient digestion [38], thereby improving production efficiency and health status [38,62,63]. In addition, the rumen microbiota plays an essential role in the development of the rumen during the early stages of life [11]. Early weaning disrupts the development of the rumen microbiota, leading to a less diverse microbial community and impaired fermentation [64]. This, in turn, can result in reduced nutrient absorption and growth in the young animal. Therefore, appropriate management practices are necessary to ensure the early establishment and development of a healthy rumen microbiota in early-weaned ruminants [65].

#### 3.2.2. The Bacterial Colonization at Rumen

It is debatable whether the gut microbial community colonizes prenatally or postnatally [40]. Bacterial communities found in numerous maternal-associated sources, including the colostrum, vagina, udder skin, and saliva, have been shown to colonize the gastrointestinal system of newborns within days of birth [40,66,67]. The first day, rumen bacterial communities of goat offspring may also be acquired from the intake of amniotic fluid in pregnancy [67]. Moreover, a recent study showed that the bacterial colonization of the fetal gut commences in utero [68]. In goats, colonization of the bacterial population has been shown to be age-dependent [39,69]. It takes 3 to 4 weeks for the bacterial community structure to stabilize, implying that this period is essential [16]. The rumen bacterial communities of goat kids exhibited remarkable alterations in three stages within the first two months after birth, according to recent longitudinal studies [10,67]. The three stages of rumen development and microbial colonization are the non-rumination phase, transition phase, and rumination phase [10,67,70]. From the non-ruminant to the ruminant phase, aerobic and facultative anaerobic microbial taxa are primarily replaced by anaerobic species in the gastrointestinal tract colonizers [71].

The microbiome gradually matures into a complex microbial community [19,72]. Most alpha diversity indices, including the Shannon index, observed bacteria, and Chao1 estimator, increase with age, suggesting that the microbiota in older age groups is more diverse than in earlier age groups [39]. This is similar to microbial colonization of the rumen contents in calves [14,73]. The three predominant phyla in newborn goats are Proteobacteria, *Bacteroidetes* and *Firmicutes* [27,40,66]. The abundance of *Proteobacteria* decreased quadratically with age at 7 days, but *Bacteroidetes* and *Firmicutes* increased [39,74]. Most of genera detected within *Firmicutes* and *Bacteroidetes* are anaerobes [74]. This could be related to the shift from an aerobic or facultative anaerobic environment niche occurring close to birth to an exclusively anaerobic one with the development of rumen [73,74]. The reduction in the phylum *Proteobacteria* and the increase in the phylum *Bacteroidetes* were found in rumen as a result of weaning [9]. As *Bacteroidetes* have a strong ability to degrade proteins and polysaccharides, they could enhance nutrient digestion and metabolism in the rumen [75]. *Bacteroidetes* were more reliant on solid diet intake than milk removal, reaching a consistent abundance after 7 weeks of age [76]. As the rumen bacterial communities are not only influenced by diet, but also by age, the fetal and newborn communities were dominated by species from the *Proteobacteria* while *Bacteroidetes* and *Firmicutes* were the two major phyla from weaning to adulthood [9,73]. However, it has been reported that after two weeks of age, the community no longer demonstrated large temporal variations at the phylum level, albeit the relative abundance of certain species remained variable [14,39,77]. *Firmicutes*, for example, was the dominant phylum in the rumen on the first day of life, and its members Bacillus and Lactococcus were prominent genera [67]. Following lactation, the primary bacterial phylum detected was *Bacteroidetes*, with extremely low amounts of *Bacillus* and *Lactobacillus* [67]. Additionally, at the genus level, the proportion of *Bacteroides* family was undetectable on the first day after birth but increased from 3 to 14 days of age [67]. Similarly, Jiao et al. [69] showed that *Bacteroides* surged in abundance during the first week. The bacterial biomarkers for goat kids during the non-rumination stage (i.e., 7 to 21 days) were mainly *Bacteroidetes* and its members (e.g., *Bacteroidaceae*, *Bacteroides*, and *Alistipes*) and several members of *Firmicutes* (e.g., *Lactobacillaceae*, *Lactobacillus*, and *Butyricicoccus*), which can be attributed to the dam’s milk-dominated feeding that is rich in lactose, protein, and fat [66]. In addition, Jiao et al. [69] indicated that the lactic acid bacteria in the phylum *Firmicutes*, such as *Enterococcus* and *Lactobacillus*, were found in the rumen of newborn goats. However, during the primary stages of development, *Bacteroides*, as the main genus within phylum *Bacteroidetes*, is immediately replaced by the *Prevotella* in the rumen after the provision of solid feed at three months of life in goats [9,39,67]. Thus, solid food disrupted the ruminal epithelial microbiota by selecting bacterial taxa that were more specific and were adapted to new substrates [39]. Irrespective of feeding type, relative abundances of ruminal *Prevotella*, *Fibrobacter*, *Ruminococcus*, and *Butyrivibrio* increased with age [69].

#### 3.2.3. *Methanogens* Colonization in the Rumen

*Methanogens* are important for digestion and gas production, and understanding the colonization of methanogens is important for the healthy gut microbiota and digestion in both infants and adults. Diversity within the archaea is much lower than that of bacteria, with only a few methanogenic groups being the top three most abundant active methanogens (*Methanosphaera*, *Methanobacteriaceae,* and/or *Methanobrevibacter*) [21,59]. The age-dependent tendency of alpha diversity did not show in the archaeal community of the goat’s rumen and gut [21,27]. *Methanogens* that use hydrogen as an energy source to reduce carbon dioxide or acetate to methane have a negative relationship with the oxidative condition within the rumen [78]. It has been reported that the methanogens initially colonized rumen on the first day after birth in goats [10,21]. Jiao et al. [10] showed that the archaeal copy numbers increased with age in goat kids. Moreover, irrespective of feeding type, relative abundances of ruminal *Methanobrevibacter* increased with age in goats [69]. Other studies showed that the methanogenic archaeal populations began to increase and stabilize after the starter feed intake due to the starter’s starchy components, which promote hydrogen production [21]. After weaning 40-day-old kids, the abundance of *Methanomicrobium* spp. and *Methanimicrococcus* spp. increased, while the abundance of the genus *Methanimicrococcus* decreased from 50 to 60 days and lost its dominance [21].

#### 3.2.4. Fungi and Protozoa Colonization in the Rumen

Whereas rumen bacteria and methanogens are early rumen colonizers [16], other microbiomes such as protozoa and fungi colonize the rumen later than bacteria and methanogens do [17]. This could be attributed to protozoa being highly sensitive to oxygen and requiring direct contact between young and adult animals for effective transmission [79]. Thus, ruminants are born protozoa-free, and rumen protozoa only become established after direct and continuous nose–nose contact with adult animals [79]. Protozoa can usually be found within 15 days postpartum in the rumen of young ruminants [10,40]. The genera *Entodinium* and *Epidinium* are dominant protozoa [59]. In neonatal ruminants, anaerobic fungi mainly composed of *Neocallimastix frontalis* appear in rumen samples collected at 7 days of age [10] or between 8 and 10 days after birth [40]. However, several invasive fungal pathogens, such as *Aspergillus* and *Candida*, were observed during the first week, suggesting that fungi may also play a role in developing ruminal mucosal innate immune function [67,69]. However, these pathogens declined to undetectable levels from 3 days to 14 days of life and replaced several predominant microbes with the changes in diet after 14 days of life, such as *Neocallimastix* sp. and *Orpinomyces* sp., which may be involved in the digestion of feed fiber in rumen [67]. Solid feeds play an indispensable role in the fungi and protozoa colonization as their levels surged during 28 days of life [10]. Moreover, Jiao et al. [69] showed that the relative abundances of ruminal *Neocallimastix* and *Entodinium* increased with age irrespective of feeding type. Overall, the rumen colonization of bacteria, protozoa, and fungi in animals is a complex and dynamic process that plays a vital role in the digestive system of ruminants.

### 3.3. Metabolic Functions

Kids rely on their dam’s milk in the early weeks of life because the rumen is physically and metabolically immature [80]. At this stage, the ruminal milieu does not form VFA and lacks activities of enzymes such as amylase, urease, protease, and xylanase, as well as a very low ammonia nitrogen production, implying a deficiency in fermentation ability and enzyme activities in the newborn goat kids [10]. Meanwhile, the intestinal microbiota can use milk carbohydrates such as lactose and oligosaccharides to produce a variety of metabolites, including VFA. As a result, a slight increase in acetate molar proportion in kids was observed at 14 days of age [10], indicating the slight increase in fermentation capability during the non-rumination period by the consumption of only milk [81]. Thus, the varied nutrition sources and hormonal signals during rumen development cause irreversible alterations in body composition (protein, fat, carbohydrates, minerals, vitamins, and water) and metabolic function of newborn ruminants [17,82]. Metabolic hormones from the adipose (leptin), liver (Insulin-Like Growth Factor 1), and gut (Ghrelin) act as signaling factors that regulate the activity of the gonadotropin-releasing hormone in the hypothalamus, which control appetite and feeding behavior [83,84]. Thus, animals with high levels of dietary protein and energy have greater concentrations of leptin and Insulin-Like Growth Factor 1 [85].

During growth, the goats’ feed supply is substantially altered from a high-fat milk diet to a forage- and concentrate-based diet [10]. The pattern of nutrient absorption shifts from glucose, fatty acids, and milk-derived amino acids to substances from both feed and microbial sources [35]. Rumen microorganisms ferment carbohydrates to produce VFA such acetate, propionate, and butyrate, all of which are used as energy sources in the ruminant body [86]. VFA and ketone bodies are considered the most reliable indicators of a completely functional rumen [35,80]. Thus, a significant increase was observed in VFA followed by a decline in acetate-to-propionate ratio due to the rise in starch digestion and amylolytic microbes degrading starch in weaned goats compared to goat kids fed on milk [10]. Furthermore, ammonia nitrogen concentration increased to reach its levels in adult goats along with the microbiota colonization processes [10]. The blood urea nitrogen increased in 30-days-old kids with increasing ruminal degradation of protein and ammonia production due to the increasing microbial activity [80]. Several enzymes such as amylase, xylanase, and carboxymethyl cellulose increased in 14-days-old kids even before the introduction of solid diets due to the significant role of microbial colonization, which occurs before the functional changes [10]. Overall, the rumen functional development occurs at 2 months in goats [10].

## 4. Weaning and Rumen Development of Young Goats

### 4.1. Weaning Methods

Choosing the best weaning protocol is important for minimizing weaning stress and maintaining the healthy growth of weaned kids [22,82]. Weaning is not recommended until the rumen has sufficient anatomical, physiological, and microbiological development [76]. Weaning methods include abrupt weaning, progressive weaning, skipping milk feedings, or different techniques as reviewed by Bélanger-Naud et al. [82]. Progressive weaning, which includes decreasing the milk quantity and/or the number of meals per day over the transition period, is the most suitable to wean kids with minimal digestive stress [25,28,82]. Before weaning, it has been suggested that animals should consume a sufficient amount of solid feed or both concentrates and hay [82]. Other methods are utilized to wean the goat kids, such as weaning according to the weight. Higher body weight indicates higher rumen development and consequently enhanced solid diet intake. Goat kids can be weaned as they reach 2 to 2.5 times their birth weight [82]. However, it was found that female kids of Saanen, Alpine, or Toggenburg breeds weaned at 15 kg grew faster and reached their optimal reproductive weight earlier as opposed to kids weaned at 10 kg [87].

Moreover, weaning age is one of the critical parameters for the success of early weaning due to the underdevelopment of the gastrointestinal tract and immune system [88]. However, the best weaning age of young ruminants can vary due to the animal’s diversity of feeding, management practices, and genotypes [22]. With increasing weaning age, the intestinal bacteria involved in the degradation of fiber and carbohydrates, such as *Ruminococcaceae*, *Lachnospiraceae*, and *Ruminococcus*, were increased in weaned goats, resulting in the improvement in intestinal digestion efficiency and the ability to resist stress [22]. However, it is important to note that weaning kids too late is costly and can be harmful to the development of the kid’s reticulo-rumen because it delays the stimulation of solid feed [18,89]. Late weaning will decrease the proactive life of ewes due to the longer lactation period, which could increase the necessity of ewes’ forage supply [90].

In intensive goat husbandry, abrupt and early weaning as a strategy to adapt the suckling ruminants to a diet composed of forage and concentrates is a key technology to increase the birth rate and lower the breeding cycle and production costs [22,23,24,25]. Weaning at early stages would benefit economic profitability because of the highest marketable milk [91,92]. Thus, compared to late weaning, early weaning reduces the postpartum convalescence, increases the reproductive efficiency, and promotes the development of digestive organs and survival of goats [29,93]. The advantages and disadvantages of early weaning in the rumen development of young goats will be addressed in detail in the following review.

### 4.2. Effects of Early Weaning on the Rumen Development of Young Goats

#### 4.2.1. Rumen Morphology

The rumen role in digestion depends on its mucosal structure, mainly the ruminal papilla shape and size [94]. Minimal rumen development was observed in goats solely fed on the dam’s milk [95]. Whereas the solid feed supplementation increased the ruminal VFA concentrations and promoted the growth of rumen papillae [43], the creep feed supplementation improved the rumen morphology, structure, and function, especially the surface area of the rumen papillae of goats during the pre- and post-weaning periods [44]. The early weaning and *ad libitum* supply of semi-solid concentrate diet increased the length, width, density, and surface area of rumen papillae in Malabari male kids as compared to natural suckling with green grass [42]. This could be due to the enhanced average daily dry matter intake because the early-weaned ruminants are more efficient in terms of the adaptation to the new solid diet [42]. Solid feed as an initiating agent promoted the development of the rumen epithelium (epithelium thickness and rumen papillae height and width) and concentration of rumen VFA in goat kids, by regulating the expression of proteins related to cell construction, fatty acid metabolism, signal transduction, and ketone body synthesis [50]. The early-weaned kids are more experienced in consuming solid feeds, which significantly benefits the small ruminants for better development [35,80,96]. Accordingly, Abdelsattar et al. [51] found that goats after weaning showed the highest papillae height, lamina propria, muscle layer thickness, and epithelial thickness. In addition, Carballo et al. [97] showed that lambs weaned at 4 weeks had a higher papillary epithelial thickness than lambs weaned at 6 weeks.

#### 4.2.2. Rumen Microbiota

Abruptly changing from a liquid diet into a complete solid feed can modify the composition of the gastrointestinal tract microbiota in young ruminants. By weaning, the proportion of *Proteobacteria* increased while most of the proportions of *Firmicutes* and *Bacteroidetes* were reduced [74], suggesting that weaning disturbs the rumen bacteria communities due to the sudden dietary changes [98]. Rumen microbial alterations in early-weaned goats are affected by both age and dietary factors [99]. According to several studies, the bacterial populations in the gastrointestinal system are regulated not just by diet but also by the animal’s age [39,67,100]. Other researchers have found that individual animals’ bacterial patterns have changed [14,39]. Shifts in rumen microbial community may be more significant when the rumen microbiota is less stable and relatively simple compared to a higher, diverse, and well-established microbial community, which has higher resilience and is more resistant to disturbances [101]. It has been reported that the young ruminant gut microbiota is more sensitive to dietary changes due to the immature gastrointestinal tract [99]. As a result, it has been claimed that an animal’s early dietary experiences have a greater and longer-lasting impact than those that occur later in life [102]. This suggests that changing the rumen microbial population during the early stages of rumen development could result in long-term effects, i.e., microbial programming [16,74]. Thus, the early nutritional interventions can manipulate the establishment of rumen microbiota to make rumen microbes able to efficiently digest fiber after weaning [103].

#### 4.2.3. Rumen Metabolic Function

Weaning is a metabolic event for the young ruminant due to the shift of the diet from liquid (milk/milk replacer) to solid (starter feed or grass) [104]. Early weaning can increase psychological stress and induces unfavorable changes in gut structure and function in goats [28]. For example, Liao et al. [23] showed that early weaning stress induced repression of the expression profiles in goats, including salivary secretion, bile secretion, vascular smooth muscle contraction, and calcium signaling pathways. Therefore, early weaning could cause impairment of the digestive system and growth retardation. Weaning reduced the dry matter, crude protein, and ether extract intake and digestion in lambs but increased the starch, neutral detergent fiber, and acid detergent fiber intake and digestion [105]. Jiao et al. [10] found that amylase and protease activity potentials showed a sharp decline 42 days after weaning in goats. The rumen epithelium is a unique site of interaction between the rumen microbial metabolism and the host [106]. An undeveloped rumen at weaning has a negative effect on nutrient digestion and absorption, which may cause diseases such as diarrhea and respiratory infections [10]. A higher starter diet intake and lower ruminal VFA and nitrogen concentrations were observed in abruptly weaned lambs on day 49 than suckling lambs, leading to low rumen epithelium thickness and papilla width in the weaning lambs [107]. Thus, rumen development requires both enough rumen VFAs and the physical stimulus of feed [107]. Therefore, it has been reported that throughout the early rumen development in goats, the co-development between the rumen and its microbiomes boosts the increasing ability of nitrogen metabolism in the rumen and the expression of genes involved in the immune response and antimicrobial activity [108].

### 4.3. Impact of Rearing System on the Rumen Development of Early Weaning Goats

Suitable feeding management and rearing systems should be considered when developing nutritional intervention techniques in the early stages of life. Kids in commercial dairy goat systems are often removed from their dam immediately after birth or within the first few hours of life and artificially nurtured until weaning [7,25,109]. This is due to the higher total feed costs of the traditional system [110]. Furthermore, the artificial feeding of kids early is critical to overcoming the paucity of milk provided by dams [95]. However, artificial rearing can limit the rumen microbiological colonization leading to negative health and digestive problems on the weaning process [76,89]. Suckling plays a significant role in the strength of the mother–newborn interaction [111]. As a result, having adult companion goats might help the transition from liquid to solid nutrition, thereby improving the weaning process compared with the separation from the dam [7,18,109]. Furthermore, the naturally suckled goat kids grew considerably faster than the early-weaned and the artificially reared goat kids [112]. Furthermore, Abecia et al. [113] showed that natural milk feeding via the dam improved rumen anatomic development, VFA, and acetate-to-propionate ratio, and decreased rumen pH, indicating greater concentrate intake in naturally fed kids compared to artificially reared kids with milk replacer. Accordingly, different colonization patterns were observed for different rearing systems, indicating that artificial milk feeding could jeopardize an optimal microbial gut colonization, especially for the protozoa and the bacteria concentration [113]. The early-isolated individuals may have a limited colonization of protozoa that transfers directly by saliva [40], as well as lower colonization of cellulolytic bacteria [11]. As a result, bacterial diversity indices grew with age and were higher in kids that remained with their mother than in kids in the absence of adults [114]. The lack of interaction with adult animals hampered rumen microbial growth, which had a detrimental impact on feed digestibility and production [115]. Furthermore, growing newborn goat kids in the presence of adult partners resulted in a more complex protozoal population and a more diversified bacterial community, as well as greater rumen pH, butyrate, and ammonia contents, indicating increased fibrolytic and proteolytic activities [109]. It is important to understand that artificial rearing stress can be reduced by providing a high-quality nutritional supply; thus, a rearing strategy involving semi-solid concentrate feeding significantly increased rumen development when compared to kids suckled on green grass [42].

In grazing systems, goat kids are usually reared with their dams, which could decrease their access to the concentrate feeds. Total VFA concentrations, total protease activity, the length of rumen papillae, and liquid-associated bacterial and archaeal copy numbers were lower in grazing goats compared to goats supplemented with concentrate [10]. Furthermore, there were significant decreases in total VFA concentrations and acetate proportions in rumen fluid of grazing kids compared to kids kept indoors, most likely due to the lower digestibility and less total energy in the grazing goats’ diet [116], while the alpha diversity of the rumen bacterial and archaeal communities increased in grazing goats fed with or without concentrate as compared to goats kept indoors [116], which could be due to the presence of adult partners. In the end, the ruminal development in relation to early weaning has become clear in this literature review.

## 5. Challenges for the Future

The development of rumen morphological structure, microbial contents, and metabolic functions starts after birth with both age and feeding system being the most important influences. Compared to milk-based diets, the early and progressive evolution of solid diet intake encourages the healthy transition from monogastric to ruminant livestock. Moreover, early weaning is usually used in intensive feeding systems to increase the economic profitability and the reproductive efficiency of dams and the ability of young goats to adapt to the new solid starter diet. However, the abrupt and early weaning could harm the goat kid’s digestive functions. Accordingly, the early-weaned goat kids may suffer due to the deficient rumen development compared to the naturally reared kids. Proper weaning time and new feeding strategies are recommended for the smooth transition from milk to solid stage to avoid stress. Further research should be conducted for quick rumen development and animal growth depending on nutritional feed supplements such as volatile fatty acids and growth promoters.

## Data Availability

All data presented in this study are available on request from the corresponding author.

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
