# Peer review of "Physical, Metabolic, and Microbial Rumen Development in Goat Kids: A Review on the Challenges and Strategies of Early Weaning"

_animals, 2023, doi:10.3390/ani13152420_

Round 1
Reviewer 1 Report
The rumen development process, the role of different rearing systems on rumen development and their relation to early weaning strategies were reviewed in this manuscript, and the rumen exhibits strong structural and functional development when newborns are prone to stress were also emphasized. The topic is interesting, and the content is fall into the scope of the present journal. However, there are still some questions need to be resolved before it can be published. The following is the comments.
Line 22-23 Re-clarifing the meansing of this statement.
Ling 25 Only rumen development was discussed in the article, suggesting change digestive tract to rumen.
Line 37-38 Provide source literature for the data.
Line 43-44 The contents of Table 1 do not contribute much to the review of this article, delete.
Line 51-52 The interplay should occurre between the host and the microbes rather than the bacteria. In the article, attention should be paid to the difference between the meaning and usage of the two words microbies and bacteria.
Line 76 Rumination function is also an important part of rumen function, and the formation and development process of this function should also be reviewed in this part.
Line 92-94 Some supporting data on the promotion of rumen development by physical fiber or butyric acid concentration in lamb diet should be provided here, which can provide some guidance for actual production.
Line 95-96 In this part, it should be provided when lambs develop to what period, the rumen volume accounts for 60-80% of the complex stomach volume, which is conducive to the producers to formulate a reasonable feeding plan according to these data to regulate the development of rumen function.
Line 103 This part should also be supplemented with relevant studies on rumen absorption function, so that the content is more complete.
Line 125 The word used in the section header is bacterial, while the word used in the content is microbiota, which not only includes bacterial, but also fungi and protozoa. Therefore, the content reviewed in this section and the literature sources cited should be the research on rumen bacteria.
Line 133 It would be better to state clearly what the three significant changes are.
Line 222 What does body composition mean? How to understand?
Line 255 There is no restriction on the breed of goat. Are all goats weaned at 15kg better than at 10kg? Because some goats are small and some goats are big. This should be qualified according to the literature.
Line 273 change this to the following.
Line 283-286 No causal relationship can be established between the previous sentence and the latter sentence. Is the increase in nutrient absorption due to an increase in the length, width, density and surface area of the rumen papillae, or is it due to an increase in feed intake?
Line 295 This statement is cumbersome, and it is recommended that lambs weaned at 4 weeks have a higher papillary epithelial thickness than lambs weaned at 6 weeks.
Line 302 Is Internet the right word?
Line 308 insert higher after a.
Line 316 It should be better digested fiber.
Line 319 It is difficult to understand that weaning is a metabolic process.
Line 331 delete disease.
Line 333 Here should be a restriction on muscle.
Line 381 According to the purpose of lamb feeding (such as seed use and fattening), the establishment of an adaptive feeding system for lamb weaning should be discussed in terms of weaning diet nutrition and structure, weaning age, and the challenges faced during weaning.
Reference
The abbreviations of journal names in the references are not in the same format.
Author Response
Manuscript Number: animals-2470066 
Dear Editor and Reviewers:
Thank you very much for your careful review and constructive suggestions about our manuscript. Those comments are helpful for authors to revise and improve the quality of our paper. We studied comments carefully, tried our best to revise and improve the manuscript, and made great changes in the manuscript according to the referees’ good comments. The modified portion is marked with tracked changes. The main corrections in the paper and the responds to the reviewer’s comments are as follows:
Line 22-23 Re-clarifing the meansing of this statement.
Response 1: This statement has been modified as suggested.
See L 24-28: This review emphasized strong structural and functional changes related to rumen development in newborn goats which often happens at weaning.
Ling 25 Only rumen development was discussed in the article, suggesting change digestive tract to rumen.
Response This has been modified as suggested.
Line 37-38 Provide source literature for the data.
Response This has been modified as suggested.
Line 43-44 The contents of Table 1 do not contribute much to the review of this article, delete.
Response This has been modified as suggested.
Line 51-52 The interplay should occurre between the host and the microbes rather than the bacteria. In the article, attention should be paid to the difference between the meaning and usage of the two words microbies and bacteria.
Response The bacteria has been replaced with microbes.
Line 76 Rumination function is also an important part of rumen function, and the formation and development process of this function should also be reviewed in this part.
Response This has been improved as suggested.
See L 164-173, Daily, goats spend much time eating and ruminating [55]. Rumination is a complex process involving regurgitation and chewing of previously swallowed feed. Lickliter [56] reported that kids start mouthing soil and grasses during their first week following birth but ruminating is first observed during week 3 of age. Before weaning, kids ruminatefor over 3 hours/day [25]. After weaning, removal of dietary milk results in a dramatic increase in the time spent for rumination (5.2 hours/day) [25], indicating the major roles of age and feeding [25,55,56]. Finally, it has been reported that the adult goats spend 7.2 hours/day reaching about one third of their day for rumination [57,58].
Line 92-94 Some supporting data on the promotion of rumen development by physical fiber or butyric acid concentration in lamb diet should be provided here, which can provide some guidance for actual production.
Response This has been improved as suggested.
See L 125-173, Htoo, et al. [44] showed that the huge particle size of roughage and content of fiber increased rumen wall by physical stimulation, and consequently enhanced rumen motility, muscularization and rumen volume in goat kids with free access to creep feed with roughage. In addition, Malhi, et al. [45] reported that intraruminal infusion of sodium butyrate at 0.3 g/kg of body weight for 28 days in goats improved the rumen papillae size, density, and surface areaThe ruminal infusion of butyrate at 0.3 g/kg of body weight in goats increased the full weight of rumen as percent of total stomach weight [46]. In addition, infusion of sodium butyrate at 0.3 g/kg of body weight can improve ruminal epithelial growth by modulating both proliferative and apoptotic genes, i.e., Bax, caspase 3, and caspase 9 [47]. Moreover, sodium butyrate supplementation at 3.2% of diet increased rumen epithelium thickness in growing rams [48]. Accordingly, the transition period should be considered by goat nutritionist for the adaptation of rumen papillae to the dietary changes.
Line 95-96 In this part, it should be provided when lambs develop to what period, the rumen volume accounts for 60-80% of the complex stomach volume, which is conducive to the producers to formulate a reasonable feeding plan according to these data to regulate the development of rumen function.
It should be in the mature goats after 2 months of age. This has been modified as suggested in the text.
Line 103 This part should also be supplemented with relevant studies on rumen absorption function, so that the content is more complete.
Response This has been modified as suggested .
See L 150-163, In addition, the development of rumen absorption function in goat kids after birth is a crucial process that enables animals to adapt to solid diets. Initially, the rumen is a small pouch-like structure with a thin lining and lacks the necessary capacity to perform efficient absorption of nutrients. However, the rumen gradually expands due to the induced cell proliferation and differentiation as the kid starts to consume solid feed [50,51]. Papillae can greatly enhance the absorptive capacity of the rumen by increasing its surface area [45,52]. High-fiber diets induce the expression of genes related to short-chain fatty acid absorption in the rumen epithelium of goats [53]. Similar improvement of absorption capacity of rumen epithelium was observed in sheep [52] and cows [54], even with unchanged absorptive surface area of the rumen papillae [54]. On the other hand, low-fiber diets can cause insufficient rumen development and impede proper nutrient digestion and absorption [35]. Therefore, it is important to carefully evaluate the impact of age and diets in further studies to determine their effectiveness in enhancing the absorptive capacity of the rumen
Line 125 The word used in the section header is bacterial, while the word used in the content is microbiota, which not only includes bacterial, but also fungi and protozoa. Therefore, the content reviewed in this section and the literature sources cited should be the research on rumen bacteria.
Response This has been modified as suggested and the proper terms were used.
Line 133 It would be better to state clearly what the three significant changes are.
Response This has been modified as suggested.
See L 205-207, The three stages of rumen development and microbial colonization are non-rumination phase, transition phase and rumination phase [10,67,70].
Line 222 What does body composition mean? How to understand?
The type of nutrients affects the animal body composition of protein, fat, carbohydrates, minerals, vitamins, and water, which in turn influences the rate of body organs growth.
Line 255 There is no restriction on the breed of goat. Are all goats weaned at 15kg better than at 10kg? Because some goats are small and some goats are big. This should be qualified according to the literature.
Response This has been modified as suggested and the breed of animals was added.
See L 343-345, However, it was found that female kids of Saanen, Alpine or Toggenburg breeds, weaned at 15 kg grew faster and reached their optimal reproductive weight earlier as opposed to kids weaned at 10 kg [87].
Line 273 change this to the following.
Response This has been modified as suggested
Line 283-286 No causal relationship can be established between the previous sentence and the latter sentence. Is the increase in nutrient absorption due to an increase in the length, width, density and surface area of the rumen papillae, or is it due to an increase in feed intake?
Response the increase in nutrient absorption should be due to an increase in the length, width, density and surface area of the rumen papillae , while the enhanced rumen papillae was due to the enhanced feed intake of semi-solid concentrate diet.This has been modified as suggested
See L 376-379, This could be due to the enhanced average daily dry matter intake because the early weaned ruminants are more efficient in terms of the adaptation to the new solid diet [42].
Line 295 This statement is cumbersome, and it is recommended that lambs weaned at 4 weeks have a higher papillary epithelial thickness than lambs weaned at 6 weeks.
Response This statement has been modified as suggested.
See L 386-391, In addition, Carballo, et al. [97] showed that lambs weaned at 4 weeks had a higher papillary epithelial thickness than lambs weaned at 6 weeks.
Line 302 Is Internet the right word?
Response Ok, the word network has been replaced with rumen bacteria communites as suggested.
Line 308 insert higher after a.
Response This has been modified as suggested.
Line 316 It should be better digested fiber.
Response This has been modified as suggested.
Line 319 It is difficult to understand that weaning is a metabolic process.
It is due to the dietary changes from liquid to solid feeds and therfore the volatile fatty aacids instead of glucose as the energy source for rumen development.
Line 331 delete disease.
Response This has been modified as suggested.
Line 333 Here should be a restriction on muscle.
It was the rumen and it was been modified as suggested.
Line 381 According to the purpose of lamb feeding (such as seed use and fattening), the establishment of an adaptive feeding system for lamb weaning should be discussed in terms of weaning diet nutrition and structure, weaning age, and the challenges faced during weaning.
The best feeding stratiges for goats to decrease weaning stress was written and submitted to a journal.
Reference: The abbreviations of journal names in the references are not in the same format.
Response This has been modified as suggested.
Reviewer 2 Report
It is a review article and is quite detailed. It is a relevant theme and there are not many articles that describe the theme in this organized and systematized way. The references are appropriate and the theme is of interest to the veterinary medical community and professionals who work in animal production.
We sent some comments in order to improve the article:
- What is the relevance of table 1 to this work?
- Is the objective to compare species or weaning times and methodologies?
- Horses and pigs are not easily comparable to ruminants
- Furthermore, colonization of the stomach with symbiotic and commensal bacteria has been shown to be critical for the establishment and maintenance of the neonate's immune system [18] - in which species?
- goat industry - replace with goat production
- Discuss what is different about goats compared to other ruminants in this respect?
- What are the most common causes of death or associated with early weaning?
- what are the authors' views on possible strategies?
- What differences are expected to be found, according to the type of intensive/extensive production?
- Could it have importance in autochthonous breeds?
- Refer more to the economic impact?
I enjoyed the review but it needs to be done in a more critical and personal way.
Congratulations on the work.
Author Response
Manuscript Number: animals-2470066 
Dear Editor and Reviewers:
Thank you very much for your careful review and constructive suggestions about our manuscript. Those comments are helpful for authors to revise and improve the quality of our paper. We studied comments carefully, tried our best to revise and improve the manuscript, and made great changes in the manuscript according to the referees’ good comments. The modified portion is marked with tracked changes. The main corrections in the paper and the responds to the reviewer’s comments are as follows:
Reviewer 2:
- What is the relevance of table 1 to this work?
It as been deleted.
- Is the objective to compare species or weaning times and methodologies?
It was the time of weaning and weaning methodologies.
- Horses and pigs are not easily comparable to ruminants
It as been removed.
- Furthermore, colonization of the stomach with symbiotic and commensal bacteria has been shown to be critical for the establishment and maintenance of the neonate's immune system [18] - in which species?
It was in the neonatal calves and it has been modified in the text.
- goat industry - replace with goat production
Response This has been modified as suggested.
- Discuss what is different about goats compared to other ruminants in this respect?
We mentined the common rearing systems of goats.
See L 74-75, Several goat rearing practices are being followed including extensive rearing, semi-intensive and intensive.
- What are the most common causes of death or associated with early weaning?
Early weaning is associated with increased risks of diarrhea and lower respiratory tract infection leading to the increase of death.
- what are the authors' views on possible strategies?
This has been added
See L 92-94, Consequently, in intensive feeding systems, early weaning requires strict management practices and deep attention [29].
- What differences are expected to be found, according to the type of intensive/extensive production?
In extensive production systems, goats kids are usually kept with their dams causing long lactation period. While doat kids in the intensive production sydtems are usually weaned early to increase the marketable milk.
- Could it have importance in autochthonous breeds?
Yes, it is important as it has been reported.
- Refer more to the economic impact?
Response This has been modified as suggested.
See L 81-84, Therefore, early weaning technology has become a common practice for goat breeding in many countries because it can reduce the production costs and increase the birth rate of the dam and the marketable milk as well as the adaptation of young ruminants to the new solid diet [22-25].
Reviewer 3 Report
Rumen development after birth and strategies and challenges for early weaning in goat kids
Mahmoud M. Abdelsattar 1,2 , Wei Zhao 1,3, Atef M. Saleem 2 , Ahmed E. Kholif 4 , Einar Vargas-Bello-Pérez 5,6 and 4 Naifeng Zhang 1,*
This paper is a review summarising physical, metabolic and microbial development of the ruminal development of young goat kids. Furthermore it tries to link these factors to feeding (mainly weaning) strategies en stress (reduction).
General
The structure of the review lacks consistency in my opinion. Also these seems to be a lack of focus. In total 101 references have be cited, but mostly not in depth (more like one sentence). Citing is mostly done in a copy-paste style, which results in marked differenced in writing style. Therefore a certain “flow” is missing.
There is no mentioning of a specific approach (M&M “like” section). What were the search terms, which fora/databases were consulted, what were inclusion/exclusion criteria to the references.
The use of capitals should be reconsidered (subheadings, keywords etc).
There is a lot of repletion of information in this paper.
Numbers under ten should be written out throughout the paper.
Think of when to use the term develop or mature.
Title: should the title include that it is a review? This might make it more attractive to potential readers. Is “after birth” a necessary fraise? Since the paper focusses on physical, metabolic and microbial development should that not be mentioned in the title? Would it be an idea to shift to : “Physical, metabolic and microbial rumen development in goat kids; challenges and strategies of early weaning. A review.”
Simple summary
Line 23: where does this stress come from? At a time young goat kids are more prone to stress, because ….. (Not just newborns, also when going to the weaning process.)
Abstract
Line 24: replace “is same with” with “similar to”
Line 26: suggestion use the term microbiome instead of microbial communities?
Lines 27-28: to solids diets. However?? It is not a contradiction.
Line 29; Where do the behavioural problems come from? Either specify or leave out. Is it relevant? It is briefly mentioned further on in the paper, but not elaborated on. If it is important give it a stage!
Line 32: in further studies
Keywords: suggestion include early weaning
Introduction
The first reference is from 1992 and this is the “anchor” of your argument? Use a more recent citation please.
Line 37: growth of income or income increases? Stock instead of stocks
Line 38: wool with should be wool of
Line 42: aren’t goats considered intermediate browsers?
Table 1
What is the relevance of this table???? None. Leave out.
In general the structure of the introduction is chosen as physical, then metabolic and finally microbial. Please be consistent with this structure throughout the paper.
Lines 47-48: digestion starts in the mouth before entering the rumen.
Line 62: describe the traditional rearing system.
Line 79 and 83: and instead of or
Line 87: what in meant by interventions?
Line 102: not just low quality but more difficult to digest.
2.2. space after dot and removal of dot at the end
Line 116: what is meant by compared to?
Line 128: However should be moreover and shown should be showed
Should there be a header for microbiome in line 137?
Line 144-149: why? Explain please? Not just sum-up facts.
It would be very helpful if the information om microbiome could be summarised in a table or figure.
2.2.3. Introduce methanogens in the introduction.
Line 186: Forty days after weaning? Or weaning of 40 day old kids?
Line 191: double that
Line 204: extra space after as
Line 221: elaborate on hormonal signals
Line 243: This should be in your introduction.
Line 250: stress or digestive stress please elaborate
Line 253: organ development? Please explain.
Lines 264-265: Why?
Line 179: breast milk or goats milk?
Line 288-297: What does this have to do with weaning age?
3.3. and disease risk reduction?
This paragraph is well written. This has flow.
At the end the relevance of ruminal development in relation to (early) weaning becomes clear. Perhaps integrate this in the paper?
Also the paper would benefit from more discussion and less summarising.
It seems there are different writing styles throughout this paper. (Might be through the literal citing and/of diffent authors.) Therfore it missies "flow".
The paper might benifit from advice from a native speaker who is knowledgable on the topic.
Author Response
Manuscript Number: animals-2470066 
Dear Editor and Reviewers:
Thank you very much for your careful review and constructive suggestions about our manuscript. Those comments are helpful for authors to revise and improve the quality of our paper. We studied comments carefully, tried our best to revise and improve the manuscript, and made great changes in the manuscript according to the referees’ good comments. The modified portion is marked with tracked changes. The main corrections in the paper and the responds to the reviewer’s comments are as follows:
Reviewer 3:
The structure of the review lacks consistency in my opinion. Also these seems to be a lack of focus. In total 101 references have be cited, but mostly not in depth (more like one sentence). Citing is mostly done in a copy-paste style, which results in marked differenced in writing style. Therefore a certain “flow” is missing. There is a lot of repletion of information in this paper. Also the paper would benefit from more discussion and less summarising.
This paper has been modified as suggested and enhanced by Professor Zhang from the Chinese Academy of Agricultural Sciences and Professor Einar Vargas-Bello-Pérez from University of the Reading.
There is no mentioning of a specific approach (M&M “like” section). What were the search terms, which fora/databases were consulted, what were inclusion/exclusion criteria to the references.
This has been added as suggested.
See L 98-106, The information search focused on studies reporting physical, metabolic, and microbial rumen development in goat kids in relation to weaning. The literature search was conducted (up to June 2023) using PubMed, Science Direct, Web of Science, Scopus, and Google Scholar databases. It summarizes only peer reviewed papers written in English language and findings from recent years concerning the rumen development in goats from neonatal to postweaning stage, whilst conference and congress contributions were excluded. While an effort was made to focus on research carried out in goat kids, we chose to include studies in other ruminant species when pertinent to the discussion.
The use of capitals should be reconsidered (subheadings, keywords etc).
We followed the journal style rearding this issue.
Numbers under ten should be written out throughout the paper.
This has been considered as suggested.
Think of when to use the term develop or mature.
This has been considered as suggested.
Title: should the title include that it is a review? This might make it more attractive to potential readers. Is “after birth” a necessary fraise? Since the paper focusses on physical, metabolic and microbial development should that not be mentioned in the title? Would it be an idea to shift to : “Physical, metabolic and microbial rumen development in goat kids; challenges and strategies of early weaning. A review.”
The title has been modified as suggested.
Simple summary: Line 23: where does this stress come from? At a time young goat kids are more prone to stress, because ….. (Not just newborns, also when going to the weaning process.)
The stress at weaning and this statement has been modified accordingly.
See L 24-28: This review emphasized strong structural and functional changes related to rumen development in newborn goats which often happens at weaning.
Abstract: Line 24: replace “is same with” with “similar to”
The has been modified as suggested.
Line 26: suggestion use the term microbiome instead of microbial communities?
The has been modified as suggested.
Lines 27-28: to solids diets. However?? It is not a contradiction.
The has been modified as suggested.
Line 29; Where do the behavioural problems come from? Either specify or leave out. Is it relevant? It is briefly mentioned further on in the paper, but not elaborated on. If it is important give it a stage!
The focus of this review was about rumen development so we deleted the extra informations about the behavioural impact of weaning.
Line 32: in further studies
The has been modified as suggested.
Keywords: suggestion include early weaning
The has been modified as suggested.
Introduction: The first reference is from 1992 and this is the “anchor” of your argument? Use a more recent citation please.
The reference been updated as suggested.
Line 37: growth of income or income increases? Stock instead of stocks
Income increases was added as suggested.
Line 38: wool with should be wool of
The has been modified as suggested.
Line 42: aren’t goats considered intermediate browsers?
Yes, it is.
Table 1: What is the relevance of this table???? None. Leave out.
It has been deleted as suggested.
In general the structure of the introduction is chosen as physical, then metabolic and finally microbial. Please be consistent with this structure throughout the paper.
Sure, we focused on these three parts in the review.
Lines 47-48: digestion starts in the mouth before entering the rumen.
Sure, It was not the first and it has been modified as suggested.
Line 62: describe the traditional rearing system.
It has been added as suggested.
See L 75-78, In traditional extensive rearing systems, goats are continuously kept in the open field compared to goats under intensive rearing systems, where they are kept indoors and stall fed.
Line 79 and 83: and instead of or
This has been modified as suggested.
Line 87: what in meant by interventions?
It means the addition of solid diet or supplementing it. I just deleted it.
Line 102: not just low quality but more difficult to digest.
It has been added as suggested….. low quality and less digestible forages…………
2.2. space after dot and removal of dot at the end
This has been modified as suggested.
Line 116: what is meant by compared to?
The bacteria has better impact than other microorganisms.
Line 128: However should be moreover and shown should be showed
This has been modified as suggested
Should there be a header for microbiome in line 137?
No, this is related to te previous subheading.
Line 144-149: why? Explain please? Not just sum-up facts.
It is related to the effect of age and the stage of rumen development. The writting style has been improved as suggested.
t would be very helpful if the information om microbiome could be summarised in a table or figure.
This can be found in the graphical abstract.
2.2.3. Introduce methanogens in the introduction.
This has been modified as suggested
See L 70-73, In addition, methanogens colonization in the rumen early life contributes to the modulation of the rumen microbiome and fermentation methane mitigation and production efficiency [21].
Line 186: Forty days after weaning? Or weaning of 40 day old kids?
It should be weaning of 40 day old kids.
Line 191: double that
This has been modified as suggested………..than bacteria and methanogens ………..
Line 204: extra space after as
This has been modified as suggested.
Line 221: elaborate on hormonal signals
This has been modified as suggested.
See L 305-309, Metabolic hormones from adipose (leptin), liver (Insulin-Like Growth Factor 1) and gut (Ghrelin) act as signaling factors that regulate activity of gonadotropin-releasing hormone in the hypothalamus which control appetite and feeding behavior [83,84]. Thus, animals with high levels of dietary protein and energy have greater concentrations of leptin and Insulin-Like Growth Factor 1 [85].
Line 243: This should be in your introduction.
This statement has been deleted from this section.
Line 250: stress or digestive stress please elaborate
It should be digestive stress
Line 253: organ development? Please explain.
It was modified to be the rumen.
Lines 264-265: Why?
Late weaned animals has less stimulation of solid feed
Line 179: breast milk or goats milk?
It has been modified to be goats milk.
Line 288-297: What does this have to do with weaning age?
The early weaned ruminants are more efficient in terms of the adaptation to the new solid diet.
3.3. and disease risk reduction?
It should be the rumen development
This paragraph is well written. This has flow. At the end the relevance of ruminal development in relation to (early) weaning becomes clear. Perhaps integrate this in the paper?
This has been modified as suggested.
See L 478-479, At the end, the ruminal development in relation to early weaning becomes clear in this literature review.
Round 2
Reviewer 1 Report
The authors have revisied the manuscript according to the commets. However, some minor changes are needed before the article can be published. The special comments is the following.
Line 93 Insert a blank between morphology and and.
Line 169 Put the serial numbers of the three references in brackets.
Line 248 Change Increase to increased.
Line 365 Move 4.2.2 to ahead of next line.
Line 421 Whether to use italics or orthography of “et al” after the author name should be consistent according to the requirements of the journal.
Line 422 Do you think the acetate to propionate ratio could reflect the fermentation capacity?
Line 423-424 This part is difficult to understand. Rewrite.
Author Response
Manuscript Number: animals-2470066 
Dear reviewer:
Thank you very much for your careful review and constructive suggestions about our manuscript. Those comments are helpful for authors to revise and improve the quality of our paper. We studied comments carefully, tried our best to revise and improve the manuscript, and made great changes in the manuscript according to the referees’ good comments. The modified portion is marked with tracked changes. The main corrections in the paper and the responds to the reviewer’s comments are as follows:
Reviewer 1:
Line 93 Insert a blank between morphology and and.
Response: This has been modified as suggested.
Line 169 Put the serial numbers of the three references in brackets.
Response: This has been modified as suggested.
Line 248 Change Increase to increased.
Response: This has been modified as suggested.
Line 365 Move 4.2.2 to ahead of next line.
Response: This has been modified as suggested.
Line 421 Whether to use italics or orthography of “et al” after the author name should be consistent according to the requirements of the journal.
Response: This has been modified as suggested.
Line 422 Do you think the acetate to propionate ratio could reflect the fermentation capacity?
Response: The acetate to propionate ratio is a measure of the metabolic pathways and balances within the microbial community during fermentation. However, the acetate to propionate ratio alone is not sufficient to fully reflect the fermentation capacity. The fermentation capacity is a complex trait determined by several factors, including the microbial community composition, environmental conditions, and substrate availability.
Line 423-424 This part is difficult to understand. Rewrite.
Response: Acetate to propionate ratio suggests that concentrate intake was greater in naturally fed kids. This has been modified as suggested.
Reviewer 2 Report
Thank you for your corrections and clarifications and I am of the opinion that the article should be published in its present form.
Author Response
Thank you very much for your careful review and constructive suggestions about our manuscript. Kindly, find the attached file of the updated manuscript.

Reviewer 3 Report
Paper can be accepted, but it still needs spell and style check.
English language and style are fine/minor spell check required.